

# Developing a GIS-based water poverty and rainwater harvesting suitability maps for domestic use in the Dead Sea region (West Bank, Palestine)

Sameer M. Shadeed [1], Tariq G. Judeh [1], Mohammad N. Almasri [2]

[1]Water and Environmental Studies Institute, An-Najah National University, Nablus, West Bank, Palestine
[2]Civil Engineering Department, An-Najah National University, Nablus, West Bank, Palestine
*Correspondence to*: Sameer M. Shadeed (sshadeed@najah.edu)

**Abstract.** In Dead Sea region as arid to semi-arid regions, water shortage and the inability to satisfy the increasing domestic water demand have been threatening the sustainable development. In such situations, domestic rainwater harvesting is considered an efficient management option to combat water poverty. This paper aims to develop a domestic water poverty (DWP) and domestic rainwater harvesting suitability (DRWHS) maps for the West Bank, Palestine (5860 km$^2$). The Analytical Hierarchy Process (AHP) together with the GIS-based weighted overly summation process (WOSP) was utilized in the development of these maps. A total of 12 and four different assessing criteria were used in the development of DWP and DRWHS maps, respectively. Results of DWP map indicate that about 57 % of the West Bank is under high to very high domestic water poverty. On the other hand, the DRWHS map indicates that about 60 % of the West Bank can be classified as high to very high suitable areas for domestic rainwater harvesting. Furthermore, DWP and DRWHS maps intersection indicates that around 31 % of the West Bank areas could be classified as high potential locations for adopting rainwater harvesting techniques for domestic purposes. Finally, the developed maps are of high value for different stakeholders to realize the importance of promoting rainwater harvesting for a self-sustaining and self-reliant domestic water supply in high water poverty areas in the Dead Sea region generally and in the West Bank particularly.

**Keywords:** Water poverty mapping, rainwater harvesting suitability mapping, domestic water supply, water resources management, AHP, GIS, Dead Sea, West Bank (Palestine).

## 1 Introduction

Water is the most influential and key limiting factor for sustainable development. In the 21st century, the most serious challenge for millions of people, worldwide, is the lack of access to safe and clean water for domestic purposes (Worm and Hattum, 2006).

In Dead Sea region, among which the West Bank (Palestine), water shortage is a dominant problem jeopardizing the sustainability of water resources for different uses (PWA, 2011). This situation became worst further due to the population growth and climate change that imposed a tremendous stress on the conventional water supplies (PWA, 2011). Furthermore, the existing political situation controls the Palestinian accessibility to their water resources (Judeh et al., 2017). In 2015, the estimated annual water supply-demand gap for domestic purposes for the entire West Bank is about 32 million cubic meters (MCM) (PWA, 2015). However, the gap will increase unless the Palestinians gain access to their available surface and ground water resources.

DWP map is a simple, straightforward and efficient tool to visualize and represent the spatial variation of domestic water poverty index (DWPI) at governorate and country levels (Thakur et al., 2014). DWP map has several pros; it gives a better understanding of the relationship between the physical availability of water, its quality and suitability for domestic use and





its accessibility. It also forms a tool for monitoring programs in the water sector and it helps in improving the situation of
communities that suffer from water poverty (van der Vyver and Jordaan, 2011).
The different areas within a governorate can be defined as water poor or water rich using water poverty index (WPI) approach.
WPI is not only limited to the physical availability of water, it also considers the social, economic, political and environmental
factors associated with water poverty (Coppin and Richards, 1990; Sullivan et al., 2003; Neupane et al., 2015). This approach
has been applied in the analysis of water stresses in many countries all over the world such as; United State of America
(James et al., 2007), India (Kher et al., 2012), South Africa, Tanzania and Sri Lanka (Mlote et al., 2002), Nepal (Thakur et
al., 2014), and West Bank (Palestine) (Isaac et al., 2008).
Generally, water-poor areas should lock for new, safe, sustainable and unconventional sources of water.  For instance,
rainwater harvesting (RWH) has deemed as a viable alternative compared with other conventional water supply options
(Abdulrazzak, 2003).
RWH is the process of collecting and storing rainwater in order to be used afterwards for different uses among which the
domestic one (Siegert, 1994; Gould and Nissen-Petersen, 1999). It is considered an ancient technology that can be dated back
to biblical times (Evenari et al., 1971; Critchley et al., 1991). In Palestine and Greece, RWH had been extensively used 4000
years ago (Evenari et al., 1971; Critchley et al., 1991).
Adopting RWH will potentially enhance the economic, environmental and social development especially in arid and semi-
arid regions under uncertainty of water supply (UNEP, 2009). The use of RWH for domestic purposes entails that water
quality is sufficiently good and within the allowable permissible limits of drinking water quality standards. Mostly, the
quality of harvested water can be controlled by proper practices (e.g. cleaning of collecting surface (roofs) and the flush away
of first storm) and simple disinfections techniques if needed (African Development Bank, 2010; Meera and Ahammed, 2018).
In arid and semi-arid regions, domestic water productivity was enhanced by adopting RWH for many years (Boers et al.,
1986; Bruins et al., 1986; Critchley et al., 1991; Abu-Awwad and Shatanawi, 1997; van Wesemael et al., 1998; Oweis et al.,
1999; Li et al., 2000; Li and Gong, 2002; Rosegrant et al., 2002; Ngigi et al., 2005; Ngigi, 2006; Oweis and Hachum, 2006;
Rockström and Barron, 2007; Mwenge Kahinda et al., 2007; Campisano et al., 2017; Singh and Turkiya, 2017; Tamaddun
et al., 2018). In the West Bank (Palestine), RWH is widely used at household level in rural areas (Shadeed, 2011). Shadeed
2011 found that approximately 40 % and 11 % of the West Bank areas are suitable and highly suitable for RWH for the
different uses, respectively. Shadeed and Lange 2010 have confirmed the possibility of RWH to bridge the deficiencies in
water supply in the Faria catchment located in the northeastern part of the West Bank.
This research aims at mapping the DWP and DRWHS maps for the entire West Bank. Thus, an integrated approach using
GIS-based MCDA was adopted. The MCDA approach entails that the choice is built on a predetermined and limited number
of decision variables (criteria) described by their attributes. Hence, the most influential criteria (layers) driving both DWP
and DRWHS mapping were identified, weighted and scored using AHP. GIS-based WOSP was then used to develop both
the DWP and DRWHS maps.
This paper is of high value as the spatial intersection between DWP and DRWHS maps were studied for the first time in the
West Bank. This in turn can help decision makers to enhance the sustainable management of the water resources in the high
domestic water poverty yet high rainwater harvesting suitable areas by adopting proper RWH techniques.



**2 Study Area**
West Bank (Palestine) is located to the west of the Dead Sea in the Middle East. It has an area of about 5,860 km².
Administratively, it is divided into 11 governorates with a total population of approximately 2.9 million (PCBS, 2017) (see
Figure 1).

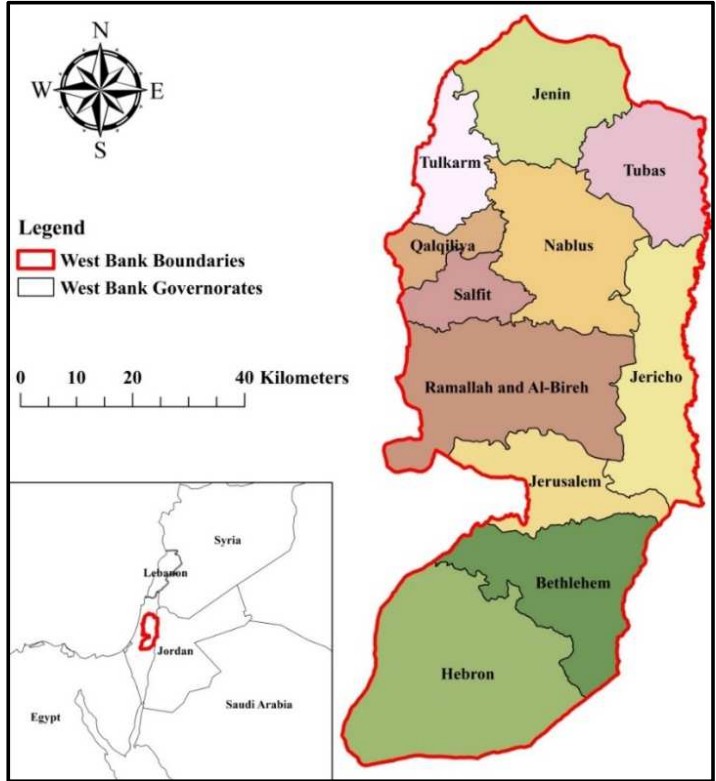


**Figure 1: Regional Location of the West Bank**
Water supply for different uses in the West Bank is very limited and does not enough to satisfy the increasing water demand.
Furthermore, the existing political situation adds another constraint on the availability and accessibility of water resources
for Palestinians. Water supply is being available either from local groundwater wells and springs or purchased from Israeli
Water Company (Mekorot). In 2015, the domestic water supply-demand gap in the West Bank was 32 MCM, whereas the
required domestic supply-demand gap (including losses) was about 65 MCM (PWA, 2015). In 2015, on ground RWH
techniques (e.g. cisterns, ponds and small scale dams) contribute to about 4 MCM for the domestic use and 3 MCM for the
agricultural use in the West Bank (PWA, 2016). According to PWA water strategy of 2018, 10 MCM is potentially collected
from adoption of different domestic and agriculture RWH techniques (PWA, 2016).
The West Bank can be classified as hot and dry during summer and cool and wet in winter (UNEP, 2003). Rainfall shows
high spatial and temporal variation, with long-term annual average rainfall of 450 mm, which is equivalent to rainfall volume
of about 2500 MCM (PWA, 2013). However, most of the annual rainfall (about 80 %) is usually occurred in winter (Shadeed,
2012). Under dry conditions, West Bank has a runoff curve number ranged from 21 to 74 with an average value of about 50
(Shadeed and Almasri, 2010). This is an indication of the high runoff potential in the country which should be utilized through
implementation of proper RWH techniques.





The land use map of the West Bank is classified into four main classes; rough grazing (62 %), agricultural practices (32 %),
built-up areas (5 %) and Israeli settlements (1 %) (MoA, 2017). Moreover, the West Bank is characterized by different soil
textures such as; clay, clay loam, loamy, sandy loam and bare rock covering 47, 31, 9, 8 and 5 % of the study area respectively
(MoA, 2017).  The elevations in the study area ranges from 375 meter below mean sea level in the vicinity of the Dead Sea
in Jericho to 1000 meter above mean sea level in the mountains of Hebron (MoP, 1997).
**3 Materials and Methods**
The overall methodological framework used in this research for developing both DWP and DRWHS maps for the entire
West Bank is illustrated in Figure 2.

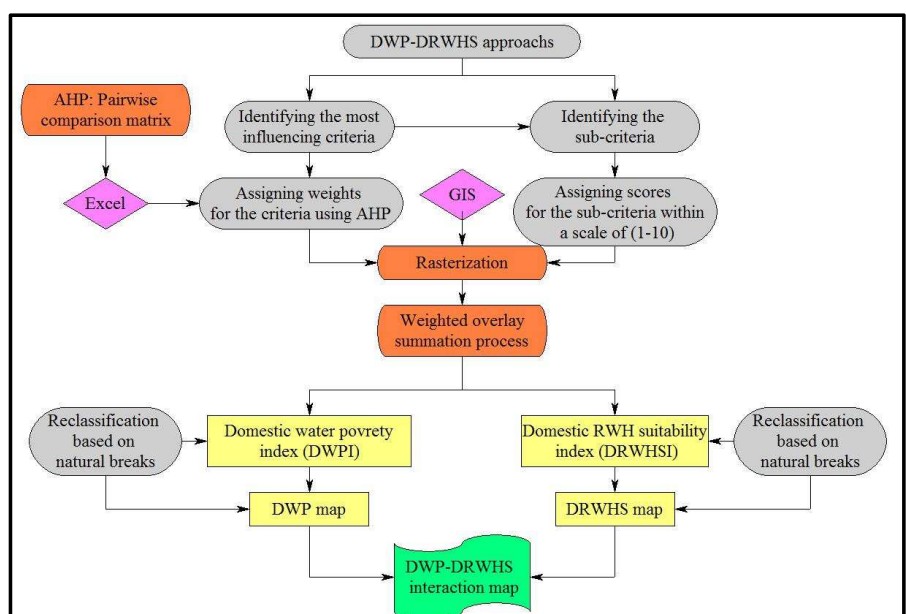

102                                              **Figure 2: methodological framework**

WPI explains water poverty in view of five key components; access, capacity, environment, resource and use (Gould and
Nissen-Petersen, 1999). In this study, the five key components were objectively represented by twelve influencing factors
(criteria) that affect the WPI in the West Bank (see Table 1). For these criteria, data were collected from different sources
which include; Palestinian Water Authority (PWA), Palestinian Central Bureau of Statistics (PCBS) and water departments
at municipalities.
The most influencing criteria on the DRWHS map in the West Bank were identified. These criteria are: rainfall depth
(RD), curve number (CN), surface slope (SS) and land use (LU). The spatial extent of the long term average annual RD
was obtained from the records of the existing rain-gauges using the inverse distance weighting method (IDW). The CN
map was developed for the entire West Bank (Shadeed and Almasri, 2010). The digital elevation model (DEM) was
processed to determine the SS layer. The LU map available at the Ministry of Agriculture (MoA) database was used.
Different weights were assigned for the different criteria used in each map by conducting the AHP pairwise comparison
matrix. The matrices were filled using a scoring system (preference values) from (1 to 9) in order to reflect the preference
and importance of the used criteria (Saaty, 1980) (See Table 2 and Table 3).





Once the pairwise comparison matrices were completed by different preference values, the AHP provides researchers the
opportunity to check and enhance the matrices consistency. However, matrices consistency was measured by estimating the
consistency ratio using the following formulas (Saaty, 1980):
$CR = \frac{CI}{RI}$
$CI = \frac{\lambda - n}{n - 1}$
Where,
*CR* consistency ratio
*CI* consistency index
*RI* random consistency index
*λ* normalized principal eigenvector
*n* number of constraints (criteria).
The matrix could be considered a consistent one if the *CR* value is smaller or equal to 0.1. Otherwise, it is considered
inconsistent and needs to be revised (Saaty, 1996, 2000). According to the different preference values used in the pairwise
comparison matrices shown in Table 2 and Table 3, the CR values for DWP and DRWHS matrices were equal to 0.04 and
0.01 respectively. So, both of them are consistent.
Each of the criteria used in DWP and DRWHS maps were divided into five sub-criteria, each of them were assigned a score
from 1 to 10 (see Table 4 and Table 5). However, DWP and DRWHS increased as the score closed to 10. Thereafter,
rasterization (cell size of 100 m by 100 m) of the different criteria based on their sub-criteria scores were manipulated by
GIS (see Figure 3 and Figure 4).





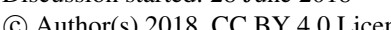

**Figure 3: The scored rasters of the 12 DWP criteria for the West Bank**





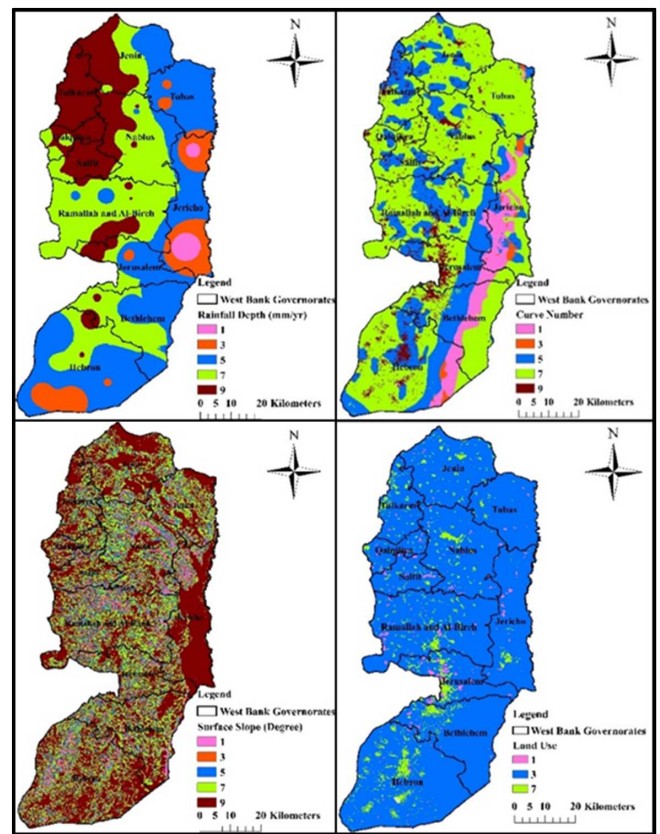

**Figure 4: The scored rasters of the four DRWHS criteria for the West Bank**
GIS is used to estimate DWPI and DRWHSI through the application of WOSP for the different layers (criteria) used. WOSP
method applies a weighted linear formula in decision-making analysis (Store and Jokimäki, 2003). It combines the different
layers based on two main parameters; the scores for the different sub-criteria for each layer and the weight of the layer itself.
However, DWPI and DRWHSI were estimated using the following formula (Malczewski, 1999):
$(\text{DWPI or DRWHSI})i = \sum_{i=0}^{n} Sij * Wj$
Where,
(DWPI or DRWHSI)i: the final DWP or DRWHS index for each cell i
Sij: the DWP or DRWHS score for each cell i in each layer j
Wj: the normalized weight for each layer j used in DWP or DRWHS maps.
**4 Results and Discussion**
**4.1 DWP Map**
Based on the methodological framework of the DWP mapping, and after performing the WOSP, the DWP map was
developed for the entire West Bank (see Figure 5). The map was classified into five water poverty categories (very low,
low, moderate, high, and very high) by natural breaks approach.





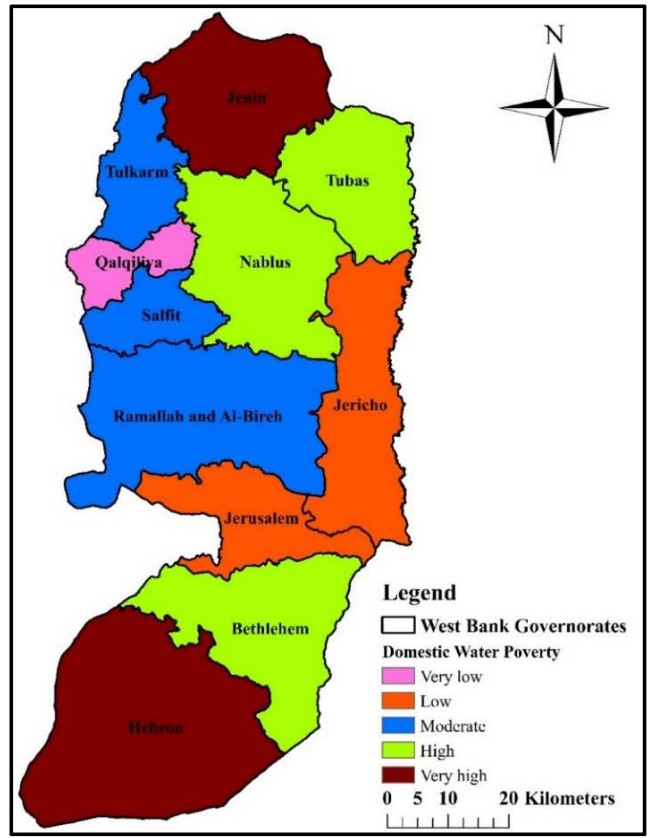

**Figure 5: DWP map for the West Bank**

Figure 5 shows that the governorates characterized by very high DWP are found to the very southern (Hebron) and very northern parts (Jenin) of the West Bank which have about 36 % of the total West Bank population (PCBS, 2017). Bethlehem, Nablus and Tubas governorates suffers from high DWP conditions. In contrast, the results indicate that Qalqiliya governorate has the lowest DWP. Whereas low to medium DWP are prevailing in the other governorates. However, the area percentages of the different DWP classes in the West Bank were estimated and illustrated in Figure 6.

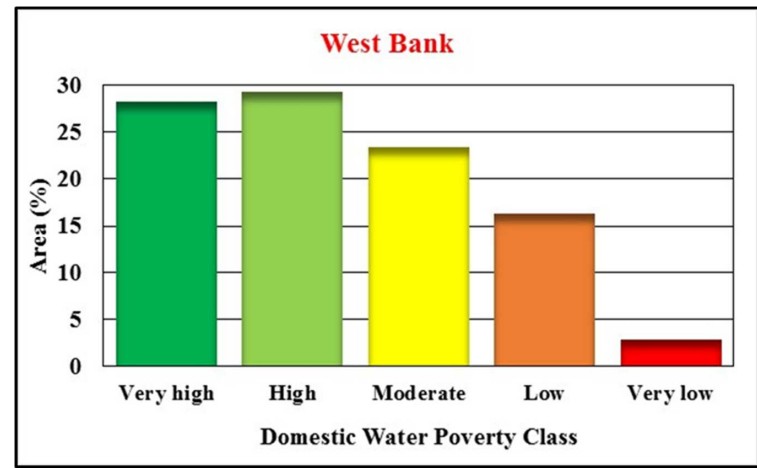

**Figure 6: Area percentages of the different DWP classes in the West Bank**



Generally, the results presented in the previous figure summarize the domestic water shortage in the West Bank. The high
to very high DWP classes form the largest area in the West Bank (approximately 57 %), and includes around 59 % of the
total West Bank population. The areas characterized by moderate DWP conditions occupy about 24 % of the total West
Bank area, and includes around 21 % of the total West Bank population. However, low to very low DWP accounts for 19
% of the total West Bank area wherein 20 % of population are living.
**4.2 DRWHS Map**
According to the methodological framework of the DRWHS mapping, and after performing the WOSP, the DWWHS map
was developed for the entire West Bank (see Figure 7). The map was classified into five suitability categories (very low,
low, moderate, high, and very high) by natural breaks approach.

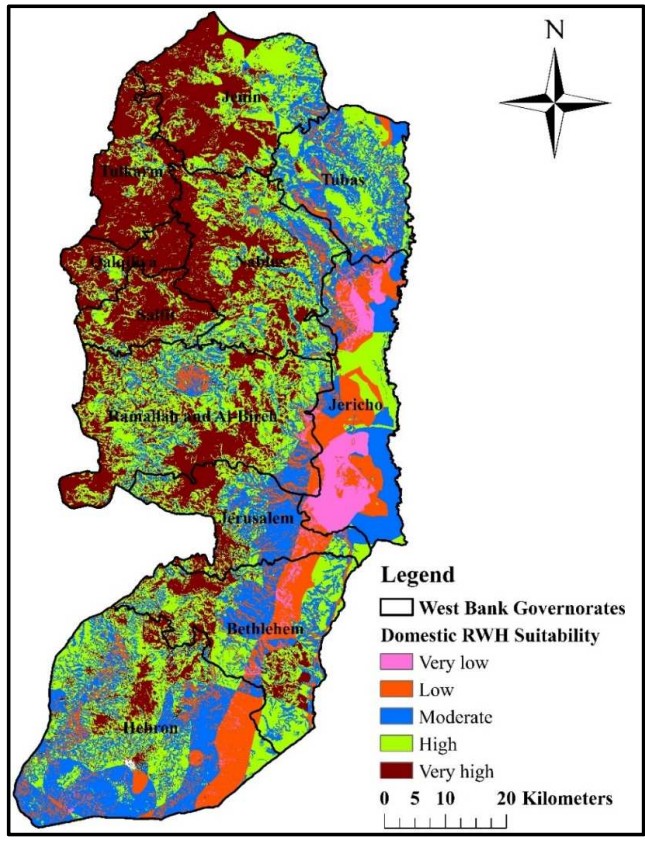

172                                **Figure 7: DRWHS map for the West Bank**

The developed DRWHS map indicates that the area characterized by very high suitable areas are distributed across the
north-western part of the West Bank, except for small portions that are located in the middle and southern mountains. In
contrast, the eastern part of the West Bank is classified as very low to low suitable areas. It is clear that the developed
DRWHS map is highly influenced by both RD and CN criteria. Where, the trend for rainfall and runoff potential increased
north-west and decreased south-east.
Generally, the area percentages of the different DRWHS classes in the different West Bank governorates are illustrated in
Figure 8.






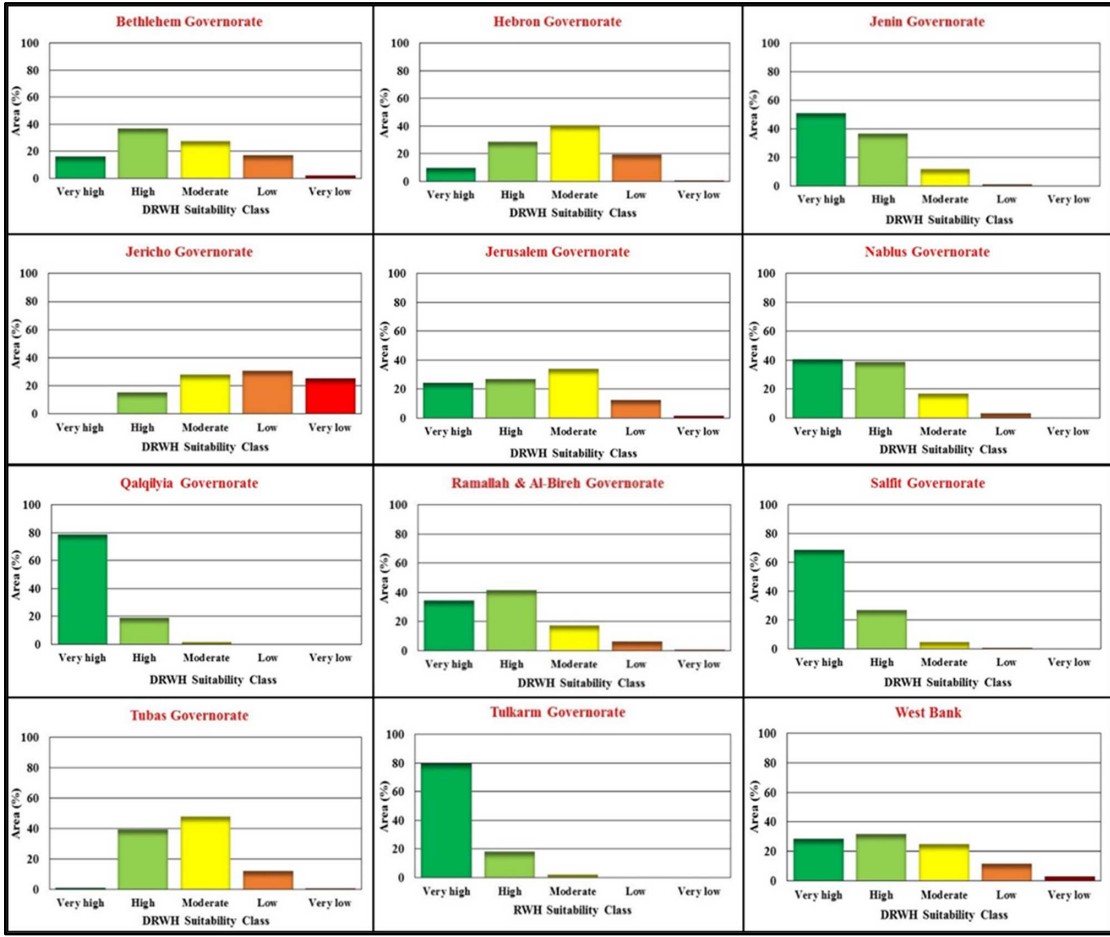

**Figure 8: Area percentages of the different DRWHS classes in the different West Bank governorates**
It's obvious that the high to very high DRWHS areas are dominant (75 %-95 %) in Qalqilyia, Tulkarm, Salfit, Jenin, Nablus
and Ramallah & Al-Bireh governorates. In general, about 60 % of the total West Bank areas are classified as high to very
high suitable for DRWH. This in turn indicates the high potential of adopting DRWH techniques in trying to bridge the
ongoing domestic supply-demand gap in the West Bank.
**4.3 DWPM-DRWHSM Intersection**
The developed DWP and DRWHS maps urged the need to identify high domestic water poverty areas yet highly suitable for
DRWH. Accordingly, spatial Intersection between both maps were conducted under the GIS environment with a special
focus on areas characterized by high to very high water poverty yet high to very high suitable for DRWH purposes. Results
are illustrated in Figure 9.





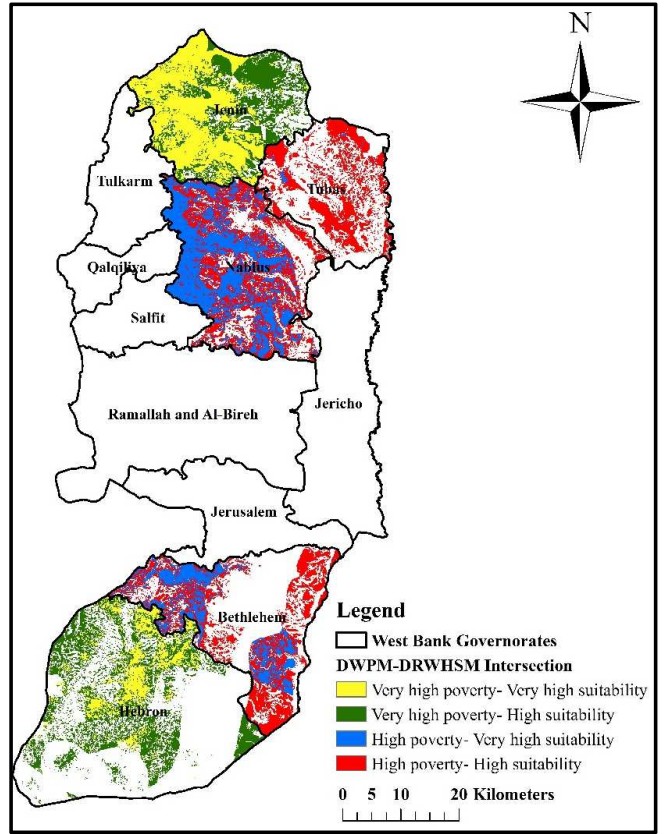

**Figure 9: DWP-DRWHS maps Intersection for the entire West Bank**

It is noticed that high to very high water poverty and rainwater suitability areas are located mostly in the northern and
southern parts of the West Bank which accounts for more than 30 % of the total West Bank area.
**5 Conclusions**
This paper came up to develop DWP and DRWHS mapping in the West Bank based on governorate scale by adopting an
integrated approach using GIS-based MCDA. Research results emphasize that the GIS-based MCDA can be used to provide
planners with a coherent and informative spatial DWP and DRWHS data. The use of MCDA for various influencing factors
is recognized to be valuable in the mapping of the DWP and DRWHS in the West Bank. Twelve and four criteria were
investigated to delineate DWP and DRWHS areas in the West Bank governorates respectively. The selection of these criteria
was affected by their impacts on DWP and DRWHS, and their availability. The AHP pairwise comparison matrix approach
was adopted to assign different criteria weights. It should be noticed that the obtained results have been subjective to the
uncertainty of the given data. Thus, it is essential to conduct a sensitivity analysis by changing the criteria weights and
criterion scores in order to quantify the severity of each one and to provide insights into the generated DWP and DRWHS
maps. The results of DWP map indicate that the high to very high DWP classes form approximately 75 % of the total West
Bank area. It is noticed that the spatial variations of DWP over different West Bank governorates are reliable and go in line
with PWA expectation. The DRWHS map indicates that there is high potential to adopt RWH especially in north-western
parts of the West Bank. Finally, although the available data are limited, the work provides an overall valuable picture about





the DWP and DRWHS in different West Bank governorates. This in turn indicates that even under data scarce regions and
limited resources yet much can be performed to assist the decision makers through providing of essential information to
mapping DWP and DRWHS areas, and thus to formulate proper strategies including the development of efficient and
comprehensive water resources management options in trying to bridge the increasing water supply-demand gap for
domestic purposes in the West Bank. The obtained results are promising to be regionalized for the entire Dead Sea region
which are facing a series water shortage challenges.
**Acknowledgements**
This work was performed within the framework of the Palestinian Dutch Academic Cooperation Program on Water
(PADUCO 2), funded by the Netherlands Representative Office (NRO) in Ramallah, Palestine. The financial support is
gratefully acknowledged.

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




**Table 1: WPI components and the associated influencing factors**

| WPI key components | Influencing factors |
|---|---|
| Access | • Time to collect water (TCW)<br>• Losses in water networks (LWN)<br>• Population served by water networks (PSWN) |
| Capacity | • Productivity (P)<br>• Citizens above poverty line (CAPL)<br>• Illiteracy (I)<br>• Average unit price of water (AUPW) |
| Environment | • Population connected to sewer networks (PCSN)<br>• Contaminated water samples by coliform (CWSC)<br>• $NO_3$ concentrations in groundwater ($NO_3$) |
| Resources | • Per capita domestic water supply (PCDWS) |
| Use | • Per capita domestic water consumption (PCDWC) |





**Table 2: AHP pairwise comparison matrix for domestic water poverty index**

| Criteria | TCW | LWN | PSWN | P | CAPL | I | AUPW | PCSN | CWSC | NO$_3$ | PCDWS | PCDWC | Weight |
|---|---|---|---|---|---|---|---|---|---|---|---|---|---|
| TCW | 1.00 | 3.00 | 2.00 | 5.00 | 5.00 | 7.00 | 2.00 | 3.00 | 0.50 | 2.00 | 4.00 | 0.50 | 0.12 |
| LWN | 0.33 | 1.00 | 0.50 | 3.00 | 4.00 | 5.00 | 0.50 | 2.00 | 0.20 | 0.50 | 2.00 | 0.17 | 0.07 |
| PSWN | 0.50 | 2.00 | 1.00 | 4.00 | 5.00 | 5.00 | 2.00 | 3.00 | 0.50 | 2.00 | 4.00 | 0.33 | 0.10 |
| P | 0.20 | 0.33 | 0.25 | 1.00 | 2.00 | 3.00 | 0.25 | 0.50 | 0.14 | 0.33 | 0.50 | 0.13 | 0.03 |
| CAPL | 0.20 | 0.25 | 0.20 | 0.50 | 1.00 | 2.00 | 0.25 | 0.33 | 0.14 | 0.25 | 0.50 | 0.13 | 0.02 |
| I | 0.14 | 0.20 | 0.20 | 0.33 | 0.50 | 1.00 | 0.20 | 0.25 | 0.13 | 0.20 | 0.33 | 0.11 | 0.01 |
| AUPW | 0.50 | 2.00 | 0.50 | 4.00 | 4.00 | 5.00 | 1.00 | 3.00 | 0.33 | 2.00 | 3.00 | 0.25 | 0.09 |
| PCSN | 0.33 | 0.50 | 0.33 | 2.00 | 3.00 | 4.00 | 0.33 | 1.00 | 0.20 | 0.50 | 2.00 | 0.17 | 0.05 |
| CWSC | 2.00 | 5.00 | 2.00 | 7.00 | 7.00 | 8.00 | 3.00 | 5.00 | 1.00 | 4.00 | 6.00 | 0.50 | 0.18 |
| NO$_3$ | 0.50 | 2.00 | 0.50 | 3.00 | 4.00 | 5.00 | 0.50 | 2.00 | 0.25 | 1.00 | 3.00 | 0.20 | 0.08 |
| PCDWS | 0.25 | 0.50 | 0.25 | 2.00 | 2.00 | 3.00 | 0.33 | 0.50 | 0.17 | 0.33 | 1.00 | 0.14 | 0.04 |
| PCDWC | 2.00 | 6.00 | 3.00 | 8.00 | 8.00 | 9.00 | 4.00 | 6.00 | 2.00 | 5.00 | 7.00 | 1.00 | 0.21 |




**Table 3: The AHP pairwise comparison matrix for domestic rainwater harvesting suitability index**

| Criteria | RD | CN | SS | LU | Weight |
|---|---|---|---|---|---|
| RD | 1.00 | 1.50 | 1.50 | 2.50 | 0.35 |
| CN | 0.67 | 1.00 | 1.50 | 2.50 | 0.31 |
| SS | 0.67 | 0.67 | 1.00 | 1.50 | 0.21 |
| LU | 0.40 | 0.40 | 0.67 | 1.00 | 0.13 |





**Table 4: DWP scoring assigned for the sub-criteria**

| # | Criteria | Sub-criteria | Score | # | Criteria | Sub-criteria | Score |
|---|----------|--------------|-------|---|----------|--------------|-------|
| 1 | TCW | <6 (days/month) | 10 | 7 | AUPW | >5.2 (NIS/m$^3$) | 9 |
|   |     | 6-12 | 8 |   |   | 4.6-5.2 | 7 |
|   |     | 13-19 | 6 |   |   | 3.9-4.5 | 5 |
|   |     | 20-26 | 4 |   |   | 3.2-3.8 | 3 |
|   |     | >26 | 2 |   |   | <3.2 | 1 |
| 2 | LWN | ≥36 (%) | 8 | 8 | PCSN | ≤20 (%) | 10 |
|   |     | 31-35.9 | 7 |   |   | 21-30 | 9 |
|   |     | 26-30.9 | 5 |   |   | 31-40 | 8 |
|   |     | 21-25.9 | 4 |   |   | 41-50 | 7 |
|   |     | <21 | 2 |   |   | >50 | 5 |
| 3 | PSWN | 76-80 (%) | 6 | 9 | CWSC | 26-30 (%) | 10 |
|   |     | 81-85 | 5 |   |   | 21-25 | 8 |
|   |     | 86-90 | 4 |   |   | 16-20 | 6 |
|   |     | 91-95 | 3 |   |   | 11-15 | 4 |
|   |     | 96-100 | 1 |   |   | 6-10 | 2 |
| 4 | P | <1 (Emp/1000 c) | 9 | 10 | NO$_3$ | ≥80 (mg/l) | 10 |
|   |   | 1.0-1.4 | 7 |   |   | 60-79 | 8 |
|   |   | 1.5-1.9 | 5 |   |   | 40-59 | 6 |
|   |   | 2.0-2.4 | 3 |   |   | 20-39 | 3 |
|   |   | ≥2.5 | 1 |   |   | <20 | 1 |
| 5 | CAPL | 65.1-72 (%) | 9 | 11 | PCDWS | <80 | 10 |
|   |     | 72.1-79 | 7 |   |   | 80-119 | 8 |
|   |     | 79.1-86 | 5 |   |   | 120-159 | 7 |
|   |     | 86.1-93 | 3 |   |   | 160-199 | 5 |
|   |     | 93.1-100 | 1 |   |   | ≥200 | 2 |
| 6 | I | 4.5-5.0 (%) | 6 | 12 | PCDWC | <40 | 10 |
|   |   | 3.9-4.4 | 5 |   |   | 40-79 | 8 |
|   |   | 3.3-3.8 | 4 |   |   | 80-119 | 6 |
|   |   | 2.7-3.2 | 3 |   |   | 120-159 | 4 |
|   |   | 2.1-2.6 | 2 |   |   | ≥160 | 2 |






**Table 5: The domestic rainwater harvesting suitability scoring assigned for the sub-criteria**

| # | Criteria | Sub-criteria | Score | # | Criteria | Sub-criteria | Score |
|---|----------|--------------|-------|---|----------|--------------|-------|
| 1 | RD | 153.0-262.1 (mm) | 1 | 3 | SS | ≥24.0 | 1 |
|   |    | 262.2-371.3 | 3 |   |    | 18-23.9 | 3 |
|   |    | 371.4-480.5 | 5 |   |    | 12-17.9 | 5 |
|   |    | 480.6-589.7 | 7 |   |    | 6-11.9 | 7 |
|   |    | 589.8-699.0 | 9 |   |    | ≤5.9 | 9 |
| 2 | CN | ≤50 | 1 | 4 | LU | Israeli settlements | 1 |
|   |    | 51-60 | 3 |   |    | Forest and rough grazing | 3 |
|   |    | 61-70 | 5 |   |    | Permanent crops and irrigated farming | 3 |
|   |    | 71-80 | 7 |   |    | Arable land | 3 |
|   |    | >80 | 9 |   |    | Built-up areas | 7 |
