# Peer review of "Developing a GIS-based water poverty and rainwater harvesting suitability maps for domestic use in the Dead Sea region (West Bank,"

_Hydrology and Earth System Sciences, 2018_

## Referee Comment (RC1) · Anonymous Referee #1 · 30 Aug 2018

The paper presents an application of spatial multi-criteria decision analysis to evaluate areas of water poverty and rainwater harvesting suitability in the West Bank, using the Analytical Hierarchy Process and weighted overlay methods. These two maps are then overlayed to determine "hotspots" of high water poverty and high rainwater harvesting suitability. Overall, the paper presents an interesting case study to determine the suitability of rainwater harvesting in a water scarce region but the presentation needs to be improved for publication. In particular, the use of English is currently not adequate throughout the text. References are also not consistent and some citations are miss-

ing or inaccurate. Although the paper presents a novel method, there is no discussion comparing this approach with related work on determining suitable locations of rainwater harvesting systems, for instance using MCDA or multi-objective optimization.

Specific comment:

L35: It would be worthwhile to write the definition of DWP

L40: Sentence should be rephrased

L47: This citation does not in appear in references

L47: There has been much research conducted on the suitability of rainwater harvesting in different parts of the world in comparison with other types of water supply systems. A paragraph could be helpful to demonstrate the higher suitability of RWH in this particular area (e.g., considering rainfall, roof areas, costs, etc.)

L63: What uses?

L63: Unusual citing style

L65: The review of literature is concentrated on RWH systems in general but should also include MCDA studies applied to RWH.

L66: Acronyms need to be defined

L72: Acronyms should be used once defined

L85: Acronyms need to be defined

L92: Why is this an indication of high rwh potential?

L108: The authors should make clear of what type of rainwater harvesting they are investigating. Is the water collected from roof runoff, surface runoff or both? The type of rainwater harvesting is likely to influence the selection of suitability criteria.

L108 & L119: Formatting of text and equations is not consistent

L111: What is the spatial resolution of land use and elevation maps?

L124: These variables should be described in greater detail for readers unfamiliar with the AHP method. For example, what is the random consistency index? How is it calculated?

L131: The score is assigned based on what?

L132: Why "however"?

Fig 3: The legends are hard to read. Also, given the values are continuous, why not use a continuous color legend?

L143: Are i and j meant to be subscripts? What is n? total number of cell or number of cells in each administrative area?

Fig 6: It would be more consistent to use the same colours as in Figure 5

L196: Results could include further discussions. For example, what does this mean to policy makers or water managers? And how is this method an improvement compared to existing methods used to determine the suitability of rainwater harvesting systems

L208: What makes variations reliable?

L209: What expectation?

L308: The authors should revise the reference section. This reference is incorrect and the formatting is inconsistent.

---

## Author Comment (AC1) · 23 Sep 2018

First of all, we would like to thank you very much for your appreciated effort in reviewing our manuscript. You really went carefully throughout the entire body of the manuscript and did some valuable comments and suggestions. Most of your comment are valid and we will do our best to modify the manuscript accordingly.

Responses to the general comment: The paper presents an application of spatial multi-criteria decision analysis to evaluate areas of water poverty and rainwater harvesting

suitability in the West Bank, using the Analytical Hierarchy Process and weighted over-lay methods. These two maps are then overlayed to determine "hotspots" of high water poverty and high rainwater harvesting suitability. Overall, the paper presents an interesting case study to determine the suitability of rainwater harvesting in a water scarce region but the presentation needs to be improved for publication. In particular, the use of English is currently not adequate throughout the text. The English will be improved as much as possible in the final version References are also not consistent and some citations are missing or inaccurate. Done. Citations in the text and the reference list is already modified. Although the paper presents a novel method, there is no discussion comparing this approach with related work on determining suitable locations of rainwater harvesting systems, for instance using MCDA or multi-objective optimization. This will be considered in the final version

Responses to the specific comment: L35: It would be worthwhile to write the definition of DWP Done and will be considered in the final version L40: Sentence should be rephrased Done and will be considered in the final version L47: This citation does not in appear in references Done and will be considered in the final version L47: There has been much research conducted on the suitability of rainwater harvesting in different parts of the world in comparison with other types of water supply systems. A paragraph could be helpful to demonstrate the higher suitability of RWH in this particular area (e.g., considering rainfall, roof areas, costs, etc.) Done and will be considered in the final version L63: What uses? Will be included in the final version (e.g. domestic, agricultural and industrials) L63: Unusual citing style Modified L65: The review of literature is concentrated on RWH systems in general but should also include MCDA studies applied to RWH. Valid point and will be considered in the final version L66: Acronyms need to be defined Done L72: Acronyms should be used once defined L85: Acronyms need to be defined Valid point. Done L92: Why is this an indication of high rwh potential? The paragraph is updated and will be considered in the final version L108: The authors should make clear of what type of rainwater harvesting they are investigating. Is the water collected from roof runoff, surface runoff or both? The type

of rainwater harvesting is likely to influence the selection of suitability criteria. Both. Will be considered in the final version L108 & L119: Formatting of text and equations is not consistent Modified L111: What is the spatial resolution of land use and elevation maps? The landuse map which was used is available as a vector data and the resolution issue is not valid. Regards the elevation map (DEM). The available and used one is of 25x25 m. L124: These variables should be described in greater detail for readers unfamiliar with the AHP method. For example, what is the random consistency index? How is it calculated? Will be considered in the final version L131: The score is assigned based on what? It was done based on personal experience of the three authors. L132: Why "however"? God point. Will be modified in the final version Fig 3: The legends are hard to read. Also, given the values are continuous, why not use a continuous color legend? You are right. Legends and unified coloring scheme will be use and considered in the final version L143: Are i and j meant to be subscripts? What is n? total number of cell or number of cells in each administrative area? i and j have to be superscript. Will be modified in the final version. n is total number of cells in each administrative area? Fig 6: It would be more consistent to use the same colours as in Figure 5 Good point. A unified coloring scheme will be use and considered in the final version L196: Results could include further discussions. For example, what does this mean to policy makers or water managers? And how is this method an improvement compared to existing methods used to determine the suitability of rainwater harvesting systems Results will be further discussed in the final version. However, our ambitious is to convince key policy makers (e.g. Palestinian water authority) to turn research outputs (findings) into development outcomes for the benefit of end-users. L208: What makes variations reliable? L209: What expectation? The developed DWP map (spatial variation) is going inline (reliable) with the existing expected water shortage issues (PWA expectation) in different governorates L308: The authors should revise the reference section. This reference is incorrect and the formatting is inconsistent. Will be updated and reformatted in the final version

---

## Referee Comment (RC2) · Dr. MJPM Riksen (Referee) · 8 Nov 2018

[referee-annotated manuscript omitted]

---

## Author Response (AR1)

**Reply to Reviewer #1**

**General Comments**

First of all, we would like to thank you very much for your appreciated effort in reviewing this manuscript. You really went carefully throughout the entire body of the manuscript and did some valuable comments and suggestions. Your comments/concerns are considered in the revised version of the manuscript.

**Specific Comments**

| Comments (lines) | Action performed | Evidence (Lines) |
|---|---|---|
| 35 | DWP definition is added. | 28-29 |
| 40 | Sentence is rephrased. | 30-32 |
| 47a | The citation is added to the reference section. | 228-229 |
| 47b | Idea is discussed. | 40-43
See also (84-87) |
| 63a | Uses are clarified. | 56 |
| 63b | Style is changed. | 57-58 |
| 65 | Literature is enriched according to your comment. | 60-62 |
| 66 | Acronym "MCDA" is defined. | 60 |
| 72 | Adjusted: the old paragraph is replaced given your comment. | 68-71 |
| 85 | It is defined. | 104 |
| 92 | Idea is clarified. | 90-91 |
| 108 | Idea is clarified. | 106-108 |
| 108 & 119 | Formatting is corrected. | 108-127 |
| 111 | The available spatial resolution is added. | 112 |
| 124 | No need to have much details in the manuscript. More details are available in the reference (Saaty, 1980). | - |
| 131 | Subjectively. | 131 |
| 132 | The pronoun "However" is replaced by "For instance" | 132 |
| Figure 3 (line 135 & 136) | 1- Legend is modified (it is easy to read).
2- Discrete color legend is better in reflecting the variations in the integer scores. | 135 |
| 143 | 1- Both i and j are subscripts (their formatting is modified).
2- Definition of the variable "n" is added | 143-145 |
| Figure 6 (line 160 & 161) | Figure 6 is adjusted to have the same color coding as in Figure 5. | 157 |
| 196 | Further discussion is conducted. | 188-190 |
| 208 | The whole sentence is removed. | - |
| 209 | The whole sentence is removed. | - |

| | | |
|---|---|---|
| 308 | The references are adjusted to be correct and consistent based on the journal authors instructions. | 227-345 |

**Reply to Reviewer #2**

**Track Changes**

All track changes are accepted and appreciated.

**General Comments**

First of all, we would like to thank you very much for your appreciated effort in reviewing this manuscript. You really went carefully throughout the entire body of the manuscript and did some valuable comments and suggestions. Your comments/concerns are considered in the revised version of the manuscript.

**Specific Comments**

| Comments (lines) | Action Performed | Evidence (Lines) |
|---|---|---|
| 13 | Sentence is deleted. | - |
| 33 | Introduction section is rearranged in the light of your comment. | 21-71 |
| 34a | DWP definition is added. | 28 |
| 34b | The comment is taken into consideration. | 32 |
| 70 to 72 | The paragraph is rephrased and shifted into conclusion section. | 217-218 |
| 73 | Chapter structure is modified according to the comment. | 72-145 |
| 83 | Sentence is rephrased. | 87 |
| 132 | Sentence is rephrased. | 132-133 |
| 133 | The term "manipulated" is replaced by the term "processed". | 134 |
| 148 | Results and discussion chapter is modified based on the comment. | 146-204 |
| 150 & 151 | Sentence is rephrased according to the comment. | 148 |
| 168 & 169 | Sentence is rephrased according to the comment. | 165 |
| 185 & 186 | The answer to your question is **"Yes"**. | - |
| 198 | Conclusion is rephrased and arranged. | 205-221 |

---

## Referee Report (RR1)

[referee-annotated manuscript omitted]

---

## Author Response (AR2)

**Reply to Reviewer #2 (Dr. MJPM Riksen)**

First of all, we would like to thank you very much for your appreciated effort in reviewing the revised version of this manuscript. You really went carefully throughout the entire body of the manuscript and did some valuable comments and suggestions. Your minor comments/concerns are considered in the final revised version of the manuscript.

**Track Changes**

All track changes are accepted and appreciated.

**Specific Comments**

| Comments (lines) | Action Performed | Evidence (Lines) |
|---|---|---|
| 101 | Caption is modified | 101 |
| 197 | The word "meddle" replaced by the word "middle" | 196 |
| 214 | Clarifications and discussions are added | (105-106) and (199-205) |

[revised manuscript text omitted]
. The DWP and DRWHS combined mapping has several advantages. It is easy to use under GIS environment, it can be applied at any region in the world once the driving factors (criteria) are made available, it helps decision makers to rely on DRWH techniques as a viable water management option for the benefit of end users in water vulnerable areas. The method has some drawbacks. For instance, the accuracy of the developed map is highly influenced by the resolution and dynamic changes (e.g. urbanization) of the data. The socio-economic constraints were not considered in this study and need to be studied first in more detail for a realistic implementation of new RWH systems.

[Figure]

**Figure 11: Percentages of DRWH activates in the northern governorates of the West Bank**

**4   Conclusions**

In this paper, maps of DWP and DRWHS were developed and utilized to identify the suitable locations for the implementation of water harvesting in order to reduce water poverty. The MCDA was employed to account for the influencing criteria according to their importance in the mapping of the DWP and DRWHS. The AHP pairwise comparison matrix approach was adopted to assign the criteria weights. Results show that 57% of the West Bank is under high to very high DWP. The DRWHS map indicates that high to very high suitable areas are concentrated in the north-western parts of the West Bank. The high to very high DWP and DRWHS areas account for more than 30% of the total West Bank area which are mostly located in the northern and southern parts. Since the MCDA entails subjectivity in assigning the weights and the scores, it will be important to conduct a sensitivity analysis. This can be done by altering the weights and scores and thereafter examining the impacts on the DWP and DRWHS maps. Despite the fact that the available data are limited, this research managed to provides a novel insight towards the identification of high domestic water poor areas. This facilitates the implementation of different DRWH techniques could be successful. This implies the applicability of this research in situations where data is limited. The work furnished herein assists the decision makers to derive proper water management strategies to bridge the gap between the supply and the demand in the West Bank. The obtained results are promising to be regionalized for the entire Dead Sea region that undergoes serious water shortage challenges. It is good to consider other spatial analysis levels for the development of the maps like the watershed outlines. Finally, further research is recommended to validate the combined map over different West Bank areas.

**Acknowledgements**

This work was performed within the framework of the Palestinian Dutch Academic Cooperation Program on Water (PADUCO 2), funded by the Netherlands Representative Office (NRO) in Ramallah, Palestine. The financial support is gratefully acknowledged. ==We are also grateful to an anonymous reviewer and to Michel Riktsen who improved our==
==manuscript.==

[revised manuscript text omitted]